# Empagliflozin-Enhanced Antioxidant Defense Attenuates Lipotoxicity and Protects Hepatocytes by Promoting FoxO3a- and Nrf2-Mediated Nuclear Translocation via the CAMKK2/AMPK Pathway

**DOI:** 10.3390/antiox11050799

**Published:** 2022-04-19

**Authors:** Yangyang Wang, Yipei Ding, Pengbo Sun, Wanqiu Zhang, Qilei Xin, Ningchao Wang, Yaoyun Niu, Yang Chen, Jingyi Luo, Jinghua Lu, Jin Zhou, Naihan Xu, Yaou Zhang, Weidong Xie

**Affiliations:** 1State Key Laboratory of Chemical Oncogenomics, Shenzhen International Graduate School, Tsinghua University, Shenzhen 518055, China; yangyang19@mails.tsinghua.edu.cn (Y.W.); spb21@mails.tsinghua.edu.cn (P.S.); wnc19@mails.tsinghua.edu.cn (N.W.); zhangyo@sz.tsinghua.edu.cn (Y.Z.); 2Open FIESTA Center, Shenzhen International Graduate School, Tsinghua University, Shenzhen 518055, China; dingyp18@tsinghua.org.cn (Y.D.); zhangwq20@mails.tsinghua.edu.cn (W.Z.); xql19@mails.tsinghua.edu.cn (Q.X.); nyy19@mails.tsinghua.edu.cn (Y.N.); yang-che19@mails.tsinghua.edu.cn (Y.C.); luojy19@mails.tsinghua.edu.cn (J.L.); ljh20@mails.tsinghua.edu.cn (J.L.); xu.naihan@sz.tsinghua.edu.cn (N.X.); 3Key Lab in Health Science and Technology, Institute of Biopharmaceutical and Health Engineering, Shenzhen International Graduate School, Tsinghua University, Shenzhen 518055, China; 4Institute for Ocean Engineering, Shenzhen International Graduate School, Tsinghua University, Shenzhen 518055, China; zhou.jin@sz.tsinghua.edu.cn

**Keywords:** empagliflozin, NASH, AMPK, CAMKK2, lipotoxicity

## Abstract

Lipotoxicity is an important factor in the development and progression of nonalcoholic steatohepatitis. Excessive accumulation of saturated fatty acids can increase the substrates of the mitochondrial electron transport chain in hepatocytes and cause the generation of reactive oxygen species, resulting in oxidative stress, mitochondrial dysfunction, loss of mitochondrial membrane potential, impaired triphosphate (ATP) production, and fracture and fragmentation of mitochondria, which ultimately leads to hepatocellular inflammatory injuries, apoptosis, and necrosis. In this study, we systematically investigated the effects and molecular mechanisms of empagliflozin on lipotoxicity in palmitic acid-treated LO2 cell lines. We found that empagliflozin protected hepatocytes and inhibited palmitic acid-induced lipotoxicity by reducing oxidative stress, improving mitochondrial functions, and attenuating apoptosis and inflammation responses. The mechanistic study indicated that empagliflozin significantly activated adenosine 5’-monophosphate (AMP)-activated protein kinase alpha (AMPKα) through Calcium/Calmodulin dependent protein kinase kinase beta (CAMKK2) instead of liver kinase B1 (LKB1) or TGF-beta activated kinase (TAK1). The activation of empagliflozin on AMPKα not only promoted FoxO3a phosphorylation and thus forkhead box O 3a (FoxO3a) nuclear translocation, but also promoted Nrf2 nuclear translocation. Furthermore, empagliflozin significantly upregulated the expressions of antioxidant enzymes superoxide dismutase (SOD) and HO-1. In addition, empagliflozin did not attenuate lipid accumulation at all. These results indicated that empagliflozin mitigated lipotoxicity in saturated fatty acid-induced hepatocytes, likely by promoting antioxidant defense instead of attenuating lipid accumulation through enhanced FoxO3a and Nrf2 nuclear translocation dependent on the CAMKK2/AMPKα pathway. The CAMKK2/AMPKα pathway might serve as a promising target in treatment of lipotoxicity in nonalcoholic steatohepatitis.

## 1. Introduction

Non-alcoholic fatty liver disease (NAFLD) is the most common chronic liver disease worldwide. It is a general term for liver steatosis, inflammation, and a series of related diseases caused by non-alcoholic factors, which affects about 25% of the global population and poses great challenges to public health [1]. Among them, non-alcoholic steatohepatitis (NASH) is an inflammatory subtype of NAFLD, which is characterized by the presence of lobular inflammation and balloon-like degeneration in the liver under the condition of steatosis, and may be accompanied by fibrosis of varying degrees [2]. Damage to hepatocytes and inflammation of the liver can promote collagen synthesis and deposition, known as fibrosis, which can lead to liver cirrhosis or even hepatocellular carcinoma, where patients may need liver transplantation [3]. Thus, the prevention and treatment of NASH are of great significance.

The pathogenesis of nonalcoholic fatty liver disease is very intricate and has not been fully elucidated. The “two-hit” hypothesis proposed by Day and James argues that excessive lipid accumulation in the liver increases the susceptibility of damage to the liver, and further increases the substrates of the mitochondrial electron transport chain (ETC), which greatly promotes the production of reactive oxygen species (ROS) and amplifies the signals of oxidative stress and inflammatory injuries, ultimately leading to the hepatocyte necrosis and liver fibrosis [3,4]. Lipotoxicity, in terms of cell injuries and death caused by the accumulation of excess saturated fatty acids, such as palmitic acid (PA), plays a critical role in the development and progression of NASH.

Sodium-glucose cotransporter 2 (SGLT2) inhibitors, a class of drugs that treat diabetes by increasing urinary glucose excretion to lower blood glucose concentration, have manifested benefits in other diseases recently. In particular, SGLT2 inhibitors improve many indicators in patients with heart failure, including oxidative response, ventricular hypertrophy, and ejection fraction. Recent clinical studies have shown that SGLT2 inhibitors can also benefit NAFLD patients, including improved lipid accumulation in the liver, ballooning degeneration, fibrosis, and liver injuries [5,6]. Moroever, study results in vivo and ex vivo have confirmed the potential of SGLT2 inhibitors in the treatment of NAFLD, but the molecular mechanisms involved remain unclear [7,8,9].

AMPKα, which is activated in response to CAMKK2, LKB1, and TAK1 signals, has been shown to regulate fatty acid oxidation and oxidative stress, and is considered to have great potential in the treatment of NAFLD [10]. Some AMPKα agonists, such as metformin, have been shown to improve NAFLD in studies. Empagliflozin (Empa), an SGLT2 inhibitor, activates AMPKα by promoting phosphorylation of AMPKα threonine172 [11]. However, the exact molecular mechanisms remained unclear. In this study, we explored the protective effects and related mechanisms of Empa in the PA-induced lipotoxicity model and confirmed the significance of CAMKK2/AMPK-regulated FoxO3a and Nrf2 co-mediated antioxidant systems in PA-induced lipotoxicity, finding that the CAMKK2/AMPK pathway may be a therapeutic target for NAFLD.

## 2. Materials and Methods

### 2.1. Cell Culture and Treatment

LO2 cells were obtained from Shanghai Cell Bank of the Chinese Academy of Sciences (Shanghai, China) and cultured in high-glucose dulbecco’s modified eagle medium (DMEM) (Gibco™, Thermo Fisher Scientific, Waltham, MA, USA) medium supplemented with 10% fetal bovine serum (Gibco™, Thermo Fisher Scientific, Waltham, MA, USA) and 1% penicillin–streptomycin antibiotics (Gibco™, Thermo Fisher Scientific, Waltham, MA, USA) in a humidity atmosphere of 5% CO_2_ at 37 °C. The cells were seeded at an appropriate density on 96-well plates, 24-well plates, 12-well plates, and 6-well plates, and then treated with 0.5 mM palmitic acid (PA; Sigma-Aldrich, St. Louis, MO, USA) for 24 h to mimic a nonalcoholic steatohepatitis model. The liquefied PA was diluted with 5% bovine serum albumin (BSA, Biofroxx, Guangzhou, China) solution gradually and sterilized by ultraviolet light to prepare a PA stock solution required by the study on the basis of a previous study [12]. Different concentrations of empagliflozin (Empa; 5.5, 11 and 22 μM; BioChemPartner, Shanghai, China), EX527 (10 μM; Beyotime, Shanghai, China), and mitoquinone (mitoQ; 500 nΜ; MedChemExpress, Shanghai, China) were added to investigate the protective effects on PA-induced LO2 cells.

### 2.2. Cell Viability

The cell viability was analyzed by Thiazolyl blue tetrazolium bromide (MTT; Sangon Biotech, Shanghai, China) assay and Calcein AM cell viability assay kit (CCK-F; Beyotime, Shanghai, China). The MTT powder was dissolved with phosphate buffer solution (PBS) and then sterilized with 0.22 μm needle filter (PALL, New York, NY, USA), and the final concentration was maintained at 5 mg/mL. After 24 h of treatment, 20 μL MTT solution was added to each well and further incubated for 4 h in the cell culture incubator. Then, the cell medium was removed, the formed formazan was dissolved with 200 μL dimethyl sulfoxide (DMSO; Sangon, Shanghai, China) solution, and the absorbance at 490 nm was analyzed by an Epoch microplate spectrophotometer (Bio-Tek, Winooski, VT, USA).

Cells were seeded into 24-well plates for CCK-F assay to analyze cell viability. The CCK-F fluorescent probe was diluted with DMEM medium at a ratio of 1:1000, adding 500 μL to each well after removing the cell medium. Then, the cells were further incubated for 20 min and finally photographed with a fluorescence microscope for observation (excitation wavelength of 494 nm; emission wavelength of 517 nm).

### 2.3. Oil Red O Staining

Intracellular lipid accumulation was analyzed by Oil red O staining. After treatment, the cells were washed twice with PBS solution, then fixed with 4% paraformaldehyde solution for 15 min. Subsequently, the cells were incubated with oil red O dye for 20 min, then quickly immersed with 60% isopropyl alcohol solution for several seconds, and following this were washed twice with PBS. Finally, 400 μL PBS solution was added to each well and they were photographed under a microscope for analysis.

### 2.4. Apoptosis Assay

For the flow cytometry assay, cells were seeded into 6-well plates. After treatment, the cell medium was collected, and then the cells were digested and centrifuged at 800 rpm for 3 min with the cell medium, incubating with Annexin V-FITC/PI apoptosis detection kit (YEASEN, Shanghai, China) reagent for 20 min. After that, the collected cells were filtered with a 200-mesh cell sieve and detected by Cytoflex (Beckman, Brea, CA, USA).

For the confocal assay, cells of suitable density were seeded into 6-well plates with cell slides and treated simultaneously. After 24 h of treatment, 4% paraformaldehyde solution was added to fix the cells, and the cells were washed with PBS once after the fixation. Finally, the slides were covered with apoptotic dye reagent, and the results were analyzed by using a confocal laser microscope.

### 2.5. LDH Release Assay

The cytotoxicity was analyzed by lactate dehydrogenase (LDH) cytotoxicity assay kit (Beyotime, Shanghai, China). Cells were seeded into 96-well plates, and the cell medium was collected after 24 h of treatment and centrifuged at 4000 rpm for 5 min. Afterward, the supernatant reacted with the prepared LDH detection solution at a ratio of 2:1 for 30 min, and then the absorbance at 490 nm was analyzed by a microplate spectrophotometer.

### 2.6. ROS Assay

The intracellular and mitochondrial ROS were detected by DCFH-DA fluorescent probe (Beyotime, Shanghai, China) and Mito-SOX^TM^ Red fluorescent probe (Thermo Fisher Scientific, Waltham, MA, USA). The cells were seeded into 24-well plates for qualitative analysis by using a fluorescence microscope (DCFH-DA, excitation wavelength of 488 nm, emission wavelength of 525 nm; Mito-SOX^TM^ Red, excitation wavelength of 510 nm, emission wavelength of 580 nm) and the detected methods, as described in our previous studies [12]. The cells were seeded into BeyoGold™ 96-well white opaque plates (Beyotime, Shanghai, China) for quantitative analysis by using a microplate reader (TECAN, Mennedorf, Switzerland), and the results were expressed as fluorescence intensity divided by cell viability.

### 2.7. Mitochondrial Functions Assay

Mitochondrial membrane potential was detected by using a mitochondrial membrane potential assay kit with JC-1 (Beyotime, Shanghai, China). For the flow cytometry assay, cells were seeded into 6-well plates. After 24 h of treatment, the cell medium and cells were collected and centrifuged at 800 rpm for 3 min and then incubated with JC-1 working solution for 20 min at 37 °C. Afterward, cells were washed with JC-1 dyeing buffer (1×) twice and filtered with a 200-mesh cell sieve, and then the fluorescence signals were detected by using Cytoflex. For the confocal assay, cells were seeded into 35 mm laser confocal culture dishes (glass bottom diameter: 20 mm) and incubated with JC-1 working solution for 20 min at 37 °C, with further washing with JC-1 dyeing buffer (1×) twice. Finally, 2 mL high-glucose DMEM medium was added into each dish, and the results were analyzed by using a confocal laser microscope.

Mitochondrial mass was analyzed by using nonyl acridine orange (NAO; MKBio, Shanghai, China) fluorescent dye. The cells were seeded into BeyoGold™ 96-well white opaque plates and incubated with 4 μM NAO working solution for 10 min after 24 h of treatment. Afterward, the fluorescence signal was analyzed by using a microplate reader (excitation wavelength of 495 nm; emission wavelength of 519 nm).

### 2.8. Cellular ATP, MDA, and SOD Assay

Cells were seeded into 6-well plates for the cellular ATP, malondialdehyde (MDA), and SOD content measurement. After treatment, cells were lysed and centrifuged at 12,000 rpm for 5 min. After centrifugation, the supernatant was collected for subsequent measurement. The cellular ATP, MDA, and SOD content were determined by the corresponding kits (Beyotime, Shanghai, China; Beyotime, Shanghai, China; Beyotime, Shanghai, China, respectively), and the protein concentration was determined by detergent compatible Bradford protein assay kit (Beyotime, Shanghai, China).

### 2.9. Intracellular Calcium Ion (Ca^2+^) Measurement

Intracellular Ca^2+^ was evaluated by the cell-permeable, calcium-sensitive fluorescent dye Fluo-3/AM (Beyotime, Shanghai, China). The cells washed once with d-HBSS buffer were incubated with 5 μM Fluo-3/AM for 25 min at 37 °C to load dye into the cells. Then, cells were washed twice with d-HBSS buffer, and the fluorescence intensity of Ca^2+^ was analyzed by fluorescence microscope and measured by ImageJ software V 1.8.0.

### 2.10. Western Blot Analysis

Cells after different treatments were lysed and collected for the preparation of protein samples. Subsequently, the samples were separated using 10% and 12.5% sodium dodecyl sulfate polyacrylamide gel electrophoresis (SDS-PAGE) gel (Epizyme Biotech, Shanghai, China), then transferred to nitrocellulose filter membranes (Pall, New York, NY, USA). After that, the membranes were blocked using 5% skim milk (Epizyme Biotech, Shanghai, China) dissolved in tris-buffered saline Tween-20 (TBST) buffer for 2 h, then incubated with primary antibodies specific to target protein diluted with 1.5% BSA prepared with TBST buffer overnight at 4 °C. After washing 3 times with TBST buffer, the membranes were incubated with secondary antibodies specific to primary antibodies for 2 h at room temperature, and then the membranes were washed again with TBST buffer. Finally, the blots were visualized using chemiluminescent horseradish peroxidase (HRP) substrate (Pierce^TM^ Western Blotting Substrate, Thermo Fisher Scientific, Waltham, MA, USA). The following primary antibodies were used in this study: β-actin (1:5000, Mouse, A1978, Sigma-Aldrich, St. Louis, MO, USA), GAPDH (1:5000, Rabbit, 10490-1-AP, Proteintech Group, Chicaog, IL, USA), SAPK/JNK (1:1000, Rabbit, 9252S, Cell Signal Technology, Boston, MA, USA), phospho-SAPK/JNK (Thr183/Tyr185) (1:1000, Rabbit, 4668S, Cell Signal Technology, Biofroxx, MA, USA), NF-κB p65 (1:1000, Mouse, 6956T, Cell Signal Technology, Boston, MA, USA), phospho-NF-κB p65 (Ser536) (1:2000, Rabbit, 3033T, Cell Signal Technology, Boston, MA, USA), Nrf2 (1:1000, Rabbit, A0674, ABclonal Technology, Wuhan, China), HO-1 (1:1000, Rabbit, A1346, ABclonal Technology, Wuhan, China), Lamin B1 (1:1000, Rabbit, 13435S, Cell Signal Technology, Boston, MA, USA), FoxO3a (1:1000, Rabbit, 12829S, Cell Signal Technology, Boston, MA, USA), phospho-FoxO3a (Ser318/321) (1:1000, Rabbit, 9465P, Cell Signal Technology, Boston, MA, USA), AMPKα (1:1000, Rabbit, 2532S, Cell Signal Technology, Boston, MA, USA), phospho-AMPKα (Thr172) (1:1000, Rabbit, 2535S, Cell Signal Technology, Boston, MA, USA), GSK3β (1:1000, Rabbit, 12456S, Cell Signal Technology, Boston, MA, USA), phospho-GSK3β (Ser9) (1:1000, Rabbit, 9323S, Cell Signal Technology, Boston, MA, USA), CPT-1α (1:1000, Rabbit, ab234111, abcam, Cambridge, UK), LKB1 (1:1000, Rabbit, A2122, Abclonal Technology, Wuhan, China), phospho-LKB1 (Ser428) (1:1000, Rabbit, AP0602, Abclonal Technology, Wuhan, China), CAMKK2 (1:1000, Rabbit, A9899, Abclonal Technology, Wuhan, China), TAK1 (1:1000, Rabbit, ET1705-14, HUABIO, Hangzhou, China), and Sirt1 (1:1000, Rabbit, A19667, ABclonal Technology, Wuhan, China). The following secondary antibodies were used to target the above primary antibodies: goat polyclonal secondary antibody to mouse IgG H&L HRP (1:5000, 7076S, Cell Signal Technology, Boston, MA, USA) and goat polyclonal secondary antibody to rabbit IgG H&L HRP (1:5000, 7074P2, Cell Signal Technology, Boston, MA, USA).

### 2.11. Elisa Assay

Cells were seeded into 12-well plates for the cellular IL-6, IL-8, TNF-α, and IL-1β content measurement to evaluate the protective effects of Empa on the inflammatory injuries in PA-treated LO2 cells. Those inflammatory cytokines were determined by using corresponding precoated ELISA kits (Dakewe Biotech Co., Ltd., Shenzhen, China) and then divided by protein concentration of cell lysates for quantitative analysis.

### 2.12. Transfection

Cells were seeded into 24-well plates, 12-well plates, or 6-well plates for transfection, and the transfection was administrated when the cell confluence reached about 70%. Lipofectamine^®^ 3000 transfection reagent (Invitrogen; Thermo Fisher Scientific, Waltham, MA, USA) and siRNA (si-FoxO3 and si-AMPKα1, RIBOBIO, Guangzhou, China) were diluted with Opti-MEM^®^ (1×, Gibco; Thermo Fisher Scientific, Waltham, MA, USA), and the final concentration of siRNA was maintained at 100 nM. Twelve hours after transfection, the cells were incubated with DMEM containing PA and Empa for another 24 h. Finally, the cells were collected for further analysis.

### 2.13. Statistical Analysis

All data in this study were presented as mean ± standard deviation (SD). The significant differences between different groups were analyzed by one-way ANOVA with Tukey’s post hoc test. It was considered that there was a significant statistical difference between data of different groups when *p* < 0.05.

## 3. Results

### 3.1. The Protective Effects of Empa on PA-Mediated Apoptosis and Injuries in LO2 Cells

PA was used to mimic a nonalcoholic steatohepatitis model in LO2 cells in this study. Firstly, we investigated the cytotoxicity of Empa and PA by MTT assay (Appendix A), and then selected 5.5, 11, and 22 μM as the experimental concentration of Empa to evaluate the protective effects of Empa in 0.5 mM PA-treated LO2 cell lines. Compared with the PA treatment group, the cells after Empa treatment showed significant increased viability (Figure 1A), and the same result was obtained from the CCK-F assay (Figure 1B). Given the truth that PA can induce hepatocyte apoptosis [13], we then investigated whether Empa also protected LO2 cells from PA-induced apoptosis. The results from confocal microscope and flow cytometry suggested that Empa administration notably improved apoptosis and necrosis (Figure 1C,D). Apoptosis and necrosis lead to structural disruption of the cell membrane, resulting in the release of intracellular enzymes into the cell culture medium, including LDH, the massive release of which is a significant marker of cell damage. To evaluate the effects of Empa on the LDH release, the LDH level in the cell culture supernatant was analyzed and the result showed that Empa significantly decreased the LDH release in contrast with the PA treatment group (Figure 1E).

Given previous studies have shown that Empa could modulate lipid accumulation in vivo and vitro [7,8], we investigated whether there was an association between Empa-mediated lipid metabolism and improved cell injuries in LO2 cells; however, Empa had no remarkable improvement in the accumulation of fatty acids (Appendix A). Taken together, these results indicated that Empa protected cells from PA-induced apoptosis and cell damage but did not modulate the whole accumulation of fatty acids in LO2 cells.

### 3.2. The Protective Effects of Empa on the Cellular and Mitochondrial ROS Production, and Mitochondrial Functions in PA-Induced Lipotoxicity Model

PA, a saturated fatty acid, induces ROS accumulation and decreases cellular ATP production in human liver cells [14]. In this study, we investigated the ROS production in LO2 cells treated with PA in the presence/absence of Empa by using the DCFH-DA probe, and the results showed that Empa could significantly suppress the massive accumulation of ROS induced by PA in LO2 cell lines (Figure 2A,B). Mitochondria are the main sites of ROS generation, especially in highly metabolized cells such as hepatocytes and cardiomyocytes. To assess the influences of PA and Empa on the mitochondrial redox homeostasis, we used the Mito-SOX probe to detect the mitochondrial-derived ROS generation [15]. The results delineated Empa remarkably reduced the mitochondrial ROS production elevated by PA and partially reversed the cell redox homeostasis (Figure 2C,D). The massive production of intracellular oxygen free radicals also causes the oxidation of lipids, proteins, and DNA. Thus, the severity of intracellular oxidative stress can also be assessed by the degree of lipid peroxidation, and the MDA measurement results also indicated that Empa administration strikingly reduced the severe oxidative stress caused by excessive PA (Figure 2I).

Mitochondrial dysfunction is also a pivotal event in PA-induced lipotoxicity, because of which we investigated the effects of Empa on mitochondrial functions in PA-treated LO2 cells. The confocal and flow cytometry results confirmed that Empa remarkably elevated mitochondrial membrane potential decreased by PA treatment (Figure 2E,F). Meanwhile, Empa intervention also showed benefit on the mitochondrial mass control. The results of NAO dyeing suggested that Empa significantly inhibited the mitochondrial fragmentation caused by PA-induced lipotoxicity (Figure 2H). Given the large amounts of mitochondria in hepatocytes, the healthy state of mitochondrial in liver cells is associated with cellular ATP generation, and the result indicated that ATP production impaired by PA was partly recovered by Empa treatment (Figure 2G).

### 3.3. Empa Protected LO2 Cells from PA-Induced Severe Inflammatory Response and Reduced p-JNK-Mediated Apoptosis

The high inflammatory level plays a significant role in the pathogenesis of NASH [16]. In this study, we investigated the effects of Empa on inflammation in the PA-induced lipotoxicity cell model. The ELISA results showed that Empa remarkably decreased IL-6, IL-8, and TNF-α levels elevated by PA, but both PA and Empa treatment had no effect on the IL-1β protein level in LO2 cells (Figure 3A–D). In addition, Western blot (WB) results illustrated that Empa administration notably inhibited the phosphorylation of p65, which was upregulated by PA (Figure 3E). Sustained JNK activation induced by saturated fatty acids plays a pivotal role in PA-induced apoptosis [17]. In this study, we verified that Empa inhibited both JNK1 and JNK2 phosphorylation remarkably, thus protecting LO2 cells from apoptosis (Figure 3F).

### 3.4. Nrf2/HO-1 and FoxO3a/SOD Pathway-Mediated Cellular Antioxidant Defense Might Be Involved in the Regulation of Empa on PA-Induced Oxidative Stress

Nrf2 is one of the most critical transcription factors regulating antioxidant defense systems, and Empa treatment significantly upregulated the protein level of total Nrf2 as well as HO-1 in LO2 cells, which is a target protein of Nrf2 (Figure 4A) [18,19]. In addition, Empa notably upregulated Nrf2 level in the nucleus of LO2 cells, while PA stimuli inhibited Nrf2 level (Figure 4B). Phosphorylated FoxO3a, which can be induced by activated AMPKα, translocated into the nucleus and ultimately enhanced the cellular antioxidant defense [20,21]. To verify whether FoxO3a was involved in the regulation of ROS by Empa, we detected the phosphorylation level of FoxO3a in LO2 cells and the expression of FoxO3a in the nucleus. Moreover, the results proved that Empa obviously upregulated the phosphorylation level of FoxO3a in LO2 cells and the level of FoxO3a in the nucleus (Figure 4C,D). Furthermore, we investigated the total SOD level in LO2 cells to evaluate the intracellular antioxidant level, which was consistent with the change of FoxO3a (Figure 4E). To further investigate whether FoxO3a was involved in the regulation of PA induced oxidative stress by Empa, si-FoxO3 was used for verification. The results of WB assays showed that FoxO3 expression in LO2 cells was significantly decreased after si-FoxO3 was added (Figure 4F). Further, FoxO3 knockdown abolished the improvement of Empa on PA-induced oxidative stress in LO2 cell lines (Figure 4G). In all, these results provided more evidence that FoxO3a was involved in the improvement of PA-induced oxidative stress in Empa.

### 3.5. Empa Activated AMPKα, GSK3β, and Sirt1 in PA-Treated LO2 Cells, Which Might Be Associated with Nrf2 and FoxO3a Signaling Axes-Mediated Antioxidant System

AMPK/GSK3β is a key regulatory pathway upstream of Nrf2 [22]. It has been reported that SGLT2 inhibitors can activate AMPK [23,24,25,26]. In view of the above, we speculated whether AMPK also mediated Nrf2 activation in the PA-induced lipotoxicity model. WB assay results confirmed that AMPKα/GSK3β pathway was significantly activated under Empa treatment (Figure 5A,B). We also studied the changes of Sirt1 protein under medication treatment and obtained the same result as *p*-AMPKα (Figure 5C).

### 3.6. Empa-Mediated Nrf2/HO-1 and FoxO3a/SOD Signaling Axes Were Dependent on AMPKα Activation

To further investigate whether Empa-mediated Nrf2/HO-1 and FoxO3a/SOD signaling axes were dependent on AMPKα activation, we transfected in LO2 cell lines with AMPKα1 siRNA. The addition of si-AMPKα1 significantly downregulated AMPKα expression in both groups treated with PA alone and those treated with PA and Empa together (Figure 6A). DCFH-DA staining showed that the ability of Empa to reduce ROS produced by PA was inhibited when AMPKα was knocked down (Figure 6B). Furthermore, WB analysis demonstrated that the ability of Empa to promote Nrf2 expression and FoxO3a phosphorylation were inhibited by si-AMPKα1 addition (Figure 6C,D). In addition, Sirt1 expression was also inhibited by si-AMPKα1 addition (Figure 6D). To investigate the effect of Sirt1 on AMPKα under Empa treatment, we used EX527, an effective and selective Sirt1 inhibitor, to inhibit Sirt1 activity. Interestingly, WB results suggested that EX527 treatment increased Sirt1 expression and the phosphorylation of AMPKα in PA-treated LO2 cells compared with the Empa group (Figure 6E). Qualitative and quantitative analysis of intracellular ROS in the setting of EX527 showed that the addition of EX527 upregulated intracellular ROS production significantly (Appendix A). Moreover, the administration of EX527 notably upregulated the phosphorylation levels of JNK1 and JNK2 (Appendix A). The inhibition of Sirt1 activity might lead to compensatory upregulation of AMPKα activity but still canceled the beneficial effects of Empa on lipotoxicity on the whole. Taken together, these results indicated that Sirt1 was not upstream of the effect of Empa on AMPKα.

### 3.7. CAMKK2, but Not LKB1 Or TAK1, Mediated AMPKα Activation by Empa Administration in PA-Treated LO2 Cells

CAMKK2, LKB1, and TAK1 are three kinases upstream of AMPKα. In order to further investigate the upstream of the ROS regulation by Empa, WB assay was used for the analysis. The results showed that Empa treatment had no effect on the phosphorylation of LKB1, although both LKB1 phosphorylation and CAMKK2 protein level were inhibited by PA stimuli, while CAMKK2 level was obviously upregulated by Empa administration (Figure 6F,G). Further, neither PA stimuli nor Empa administration had an effect on the expression of TAK1.

### 3.8. Protective Effects of ROS Inhibition by mitoQ on PA-Induced Lipotoxicity

ROS plays a key role in lipotoxicity in PA-treated liver cells. To validate whether PA-induced lipotoxicity can be attenuated by ROS elimination, we used mitoQ, a specific ROS scavenger, to inhibit the production of ROS induced by PA (Figure 7A). mitoQ significantly increased cell viability and inhibited cell apoptosis (Figure 7B,C). mitoQ also significantly inhibited the activation of p65 and JNK (Figure 7D). Considering that Empa inhibited the production of ROS induced by PA, these results indicated that ROS was really a key druggable target for lipotoxicity in liver cells and Empa might attenuate lipotoxicity, likely by controlling ROS production induced by PA.

## 4. Discussion

In the context of overload of metabolites derived from toxic lipids such as saturated fatty acids (SFA), lipotoxicity is considered as a primary cause of hepatocyte dysfunction and disease progression in NASH [27]. Excessive accumulation of toxic lipids in the metabolic process produces a large number of reactive oxygen species, causing severe oxidative stress, which in turn impairs mitochondrial functions, resulting in apoptosis, and triggers the relevant inflammatory pathways [28]. Palmitic acid (C16:0), the major membrane of saturated fatty acids in daily diets, is commonly used to mimic a lipotoxicity environment in many studies [29,30]. Here, we exposed LO2 cells to excessive palmitic acid for 24 h to construct a nonalcoholic steatohepatitis model in vitro and investigated the protective effects of Empa against hepatocellular lipotoxicity. Previous studies have found that Empa could mitigate nonalcoholic steatosis in vivo and in vitro by alleviating insulin resistance, lipogenesis, and activated autophagy [7,8,31]. Interestingly, Empa did not remarkably improve lipid accumulation in LO2 cells treated with palmitic acid in the present study, which was consistent with our previous findings in cardiomyocytes [12]. These results indicated that Empa alleviated lipotoxicity in palmitic acid-treated cells, independent of intracellular lipid accumulation. Conversely, cell viability and LDH assay elucidated that Empa protected LO2 cells from PA-induced cell damage, which suggested that Empa might mitigate the lipotoxicity of palmitic acid in other mechanisms than by regulating holistic lipid accumulation in vitro. Oversupply of saturated fatty acids can lead to incomplete fatty acid oxidation in mitochondria, increased redox pressure on the electron transport chain, and increased generation of ROS, with ensuing oxidative stress and cellular injuries [32,33,34]. Therefore, the excess of ROS is also can be a crucial candidate to evaluate the severity of lipotoxicity. Empa and dapagliflozin have been proven to have the possibility to dampen the ROS generation in mitochondria and cytoplasm ex vivo, which resembled our previous observation that canagliflozin alleviated intracellular ROS accumulation in cardiomyocytes [12,35,36]. In this study, we confirmed that Empa also exhibited beneficial effects against PA-generated ROS, particularly in mitochondria. Overproduction of ROS in mitochondria promotes mitochondrial dysfunction, including the loss of mitochondrial membrane potential, increase of mitochondrial mass, decrease of ATP production, and changes of ETC complexes [37]. Previous studies have confirmed that exogenous high-glucose and high-fat overnutrition could intensify intracellular mitochondrial depolarization and promote the increase of mitochondrial quality, which was generally manifested as abnormal division and fragmentation of mitochondria [37,38,39,40]. In a previous model of diabetes-induced myocardial microvascular injury, Empa was found to activate dynamin-relatedprotein (Drp) phosphorylation through the AMPK pathway and subsequently inhibit mitochondrial division [41]. Likewise, Empa performed analogous benefits in atrial myocytes in a high-fat/STZ-induced rat model of diabetes, e.g., improved impaired mitochondrial respiratory chains, preserved mitochondrial membrane potential, and ameliorated mitochondrial biogenesis [40]. Previous studies suggested that Empa showed protective effects on impaired mitochondrial functions, which might be one of the reasons that Empa has been approved by the FDA to improve and treat heart failure patients [42]. By further evaluating mitochondrial functions, we found that Empa administration rescued reduced mitochondrial membrane potential, recovered impaired cellular ATP production, and inhibited high fat-induced mitochondrial fragmentation in vitro. The increase in mitochondrial biogenesis under high-fat treatment may be one of the mechanisms by which cells respond to the decrease in ATP level and mitochondrial membrane potential, thus compensating for the impairment in mitochondrial functions [43].

Continuous exposure to palmitic acid and palmitate for cells induces the activation of damage-associated molecular pattern receptors, such as Toll-like receptor 4 (TLR4), which triggers phosphorylation of NF-κB and the subsequent production of inflammatory cytokines [44]. In previous studies, SGLT2 inhibitors were found to reduce the levels of serum and hepatic inflammatory factors, for instance, IL-6, TNF-α, and MCP-1 [8,26,45,46,47]. In harmony with previous studies, Empa treatment notably inhibited the activation of NF-κB and decreased the protein level of inflammatory cytokines, e.g., IL-6, IL-8, and TNF-α in LO2 cell lines in this study. Exogenous saturated fatty acid treatment can activate liver apoptosis through JNK, which is a significant feature of NASH [48]. Previous studies have shown that complete activation of JNK required both JNK1 and JNK2 [49]. Our current study proved that phosphorylation levels of both JNK1 and JNK2 activated by PA were remarkably ameliorated by Empa intervention. Further, the same results were also observed when mitoQ, a mitochondrial ROS-specific scavenger, was added, which significantly reduced ROS generation and improved the impaired cell viability and mitochondrial membrane potential by PA stimuli in LO2 cells. Meanwhile, mitoQ treatment obviously downregulated the phosphorylation levels of NF-κB and JNK. Thus, we concluded that Empa-mediated improvement of PA-induced inflammation and apoptosis depended on its clearance on ROS in PA-treated LO2 cells.

In eukaryotes, Nrf2/HO-1 is a classic antioxidant defense signaling pathway. Activation of Nrf2 inhibits oxidative stress and inflammatory injuries during treatment of NASH, which has been demonstrated in many related studies [50,51]. AMPK, an intracellular energy sensor, can directly phosphorylate Nrf2 and promote its nuclear accumulation [52]. Moreover, AMPK can inhibit GSK3β activity by facilitating the phosphorylation of GSK3β, the activation of which can catalyze the nuclear exclusion of Nrf2 [22,53]. A previous study has reported that Empa upregulated Nrf2/HO-1-mediated antioxidant responses in the hearts of C57BL/6J mice treated with a high-fat diet [23]. Here, Nrf2/HO-1 signaling axis is similarly significantly activated in PA-induced hepatocyte lipotoxicity by Empa administration. Intriguingly, the FoxO3a/SOD pathway was also extraordinarily energetic in our study. FoxO3a, a member of the forkhead transcription factor family, has been shown to be phosphorylated by AMPK to promote its nuclear translocation, which is involved in regulating oxidative stress, apoptosis, and other cellular life processes [20,54,55]. At present, there are few depictions for SGLT2 inhibitors to regulate oxidative stress, and previous studies mainly focus on delineating the role of the Nrf2/ARE-induced antioxidant defense system [56]. In this study, it was firstly proposed that Empa activates FoxO3a-mediated antioxidant pathways, providing a new explanation for SGLT2i to regulate oxidative stress. Considering that ROS is largely produced by the β-oxidation process of palmitic acid in mitochondria, Empa might also regulate oxidative stress and subsequent injuries by regulating palmitic acid metabolism in mitochondria. However, in our observation, the fatty acid transporter CPT-1α, which mediates palmitic acid into the mitochondria, was further upregulated under Empa treatment, compared with exogenous palmitic acid-mediated CPT-1α upregulation (Appendix A) [57]. Net upregulation of CPT-1α by Empa administration suggested enhanced mitochondrial fatty acid β-oxidation, which was consistent with the results of a previous study that Empa normalized mitochondrial functions in high-fat diet-impaired hearts by promoting fatty acid oxidation, suggesting that Empa might not be able to regulate oxidative stress through regulating substrates of ROS production [58].

AMPK acts as an upstream regulator of Nrf2 and FoxO3a [59]. It seems to be a consensus that SGLT2 inhibitors can activate AMPKα, especially in the case of overnutrition [23,24]. Here, we found that AMPKα was significantly activated by phosphorylation of Thr172 (pThr172) by Empa administration in the PA-induced hepatocyte lipotoxicity model. Further, the addition of si-AMPKα1 enabled us to determine that the regulation of ROS by Empa and Empa-induced nuclear accumulation of Nrf2 and FoxO3a depended on the phosphorylation of AMPKα. Sirt1, a nicotinamide adenine dinucleotide (NAD)-dependent deacetylase, activates AMPKα by deacetylating LKB1 and is also closely related to FoxO3a [60]. Given that Empa significantly upregulated the expression of Sirt1 in this study, we speculated whether Sirt1 was upstream of the effect of Empa on AMPKα. Interestingly, the addition of EX527 upregulated AMPKα phosphorylation levels compared with the combined treatment group of PA and Empa. However, this was not the first time such a result has been reported. At least in another study, EX527 also upregulated AMPKα phosphorylation levels in INS-1 cells treated with high glucose [61]. These results indicated that Sirt1 was not upstream of the effect of Empa on AMPKα. In mammals, CAMKK2, LKB1, and TAK1 have been identified as three upstream kinases that activate AMPKα Thr172 phosphorylation. AMP induces phosphorylation of AMPKα subunit Thr172 by the phosphorylation of LKB1, and CAMKK2 is activated by intracellular calcium concentration. Although AMP has also been reported to activate CAMKK2, this result seems controversial [62,63]. Our results indicated that both CAMKK2 and *p*-LKB1/LKB1 were downregulated by PA, but Empa had no significant effect on phosphorylation of LKB1, unlike CAMKK2, which was obviously upregulated after Empa treatment. Moreover, PA and PA + Empa treatment had no effects on the expression of TAK1 in LO2 cell lines. Similarly, in previous studies, a high-fat diet was found to significantly downregulate CAMKK level in the liver of mice, and the lipid content in hepatocytes of mice was remarkably increased when CAMKK2 was knocked out [64,65,66]. Therefore, Empa’s improvement of PA-induced hepatocyte lipotoxicity was likely to depend on its upregulation of CAMKK2 expression. Further studies showed that Ca^2+^ did not seem to mediate the upregulation of CAMKK2 by Empa, suggesting that Empa may upregulate CAMKK2 expression by other means than Ca^2+^ (Appendix A).

In addition, to investigate whether Empa itself could affect normal cell function, we investigated most effects of Empa in PA-untreated normal LO2 cells as did in PA-treated cells. Empa did not promote cell proliferation (Appendix A) and affect ROS production and MDA and SOD levels in normal LO2 cells (Appendix A–C); did not change mitochondrial membrane potential, ATP level, and mitochondrial mass in normal LO2 cells (Appendix A); did not promote AMPK and FoxO3 activities in normal LO2 cells (Appendix A); and did not affect JNK and p65 phosphorylation and Nrf2/HO-1 pathway in normal LO2 cells (Appendix A). However, Empa slightly upregulated GSK3β phosphorylation and downregulated CAMMK2 levels. These differences remain for further investigation in the future. Despite this, Empa could specifically attenuate PA-induced lipotoxicity and improve cell functions by unique mechanisms described above, but could not affect normal cell functions to the least extent.

## 5. Conclusions

In sum, our study firstly confirmed that Empa activated AMPKα-mediated FoxO3a and Nrf2-constructed antioxidant defense system via CAMKK2, protecting hepatocytes from PA-induced lipotoxicity, such as oxidative stress, apoptosis, necrosis, and inflammatory responses (Figure 8). In PA-untreated LO2 cells, Empa itself could not affect normal cell function but might specifically attenuate PA-induced lipotoxicity. Moreover, our study provided a new perspective to explain the improvement of Empa in NAFLD patients’ related indicators in clinical trials and the superiority of Empa in ameliorating lipotoxicity, which may be applicable to other SGLT2 inhibitors as well. Drug design targeting CAMKK2/AMPKα pathway will likely benefit patients with NASH, but this should need to be further validated in vivo and supported by additional related studies.

## Figures and Tables

**Figure 1 antioxidants-11-00799-f001:**
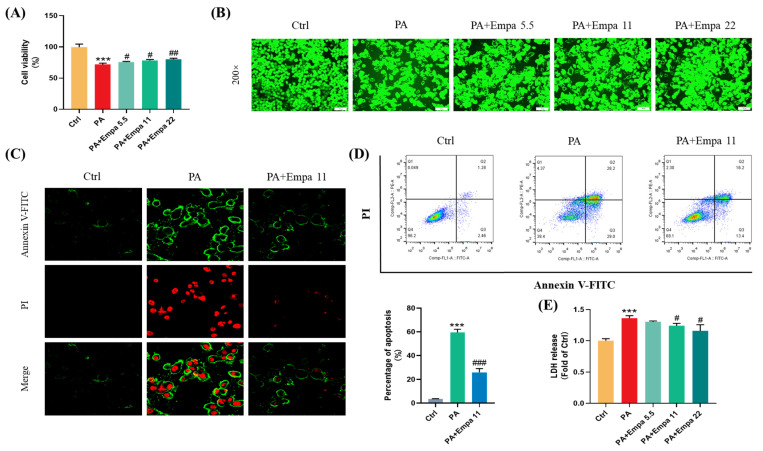
Empa improved cell viability and attenuated apoptosis of LO2 cells in a PA-induced lipotoxicity model. (**A**) Effect of Empa (5.5–22 μM) on cell viability of LO2 cells treated with 0.5 mM PA for 24 h, and which was analyzed by MTT assay. (**B**) Representative images of live cells stained with CCK-F. (**C**) Representative confocal microscope images of LO2 cells stained with Annexin V-FITC/PI. (**D**) Quantification analysis of apoptosis in LO2 cells by flow cytometry. (**E**) LDH level in the culture supernatant of LO2 cells treated with PA and Empa. All data are expressed as mean ± SD (*n* = 3). *** *p* < 0.001 versus Ctrl. # *p* < 0.05, ## *p* < 0.01, and ### *p* < 0.001 versus PA. Ctrl, PA-untreated normal control group; PA, PA-treated control group; PA + Empa 5.5–22, PA-treated and Empa-treated (at final concentrations of 5.5–22 μM) groups.

**Figure 2 antioxidants-11-00799-f002:**
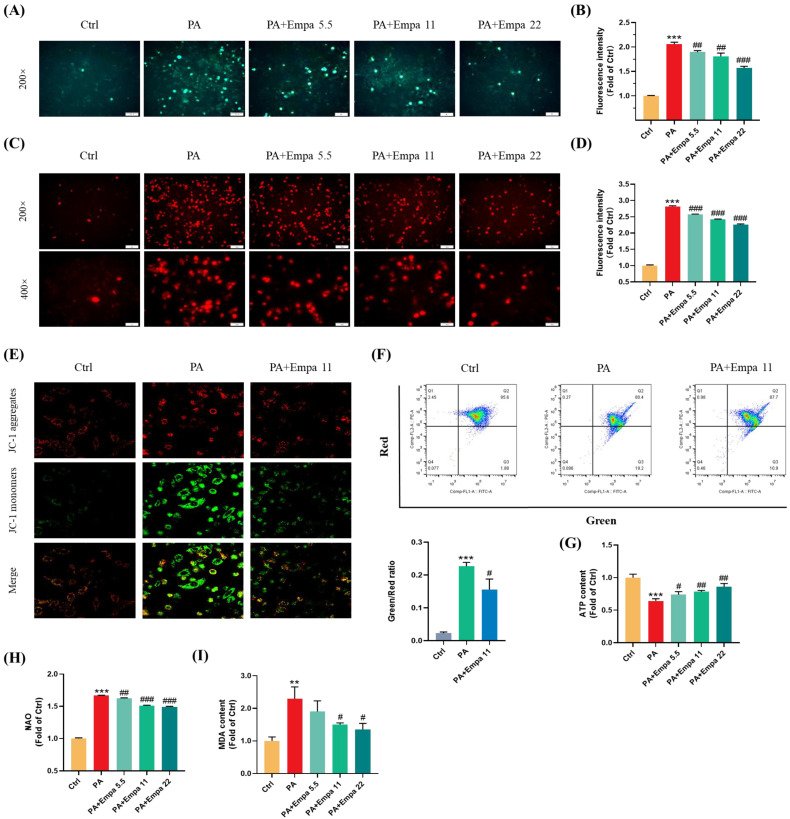
Empa attenuated mitochondrial ROS generation and improved redox homeostasis as well as mitochondrial functions in PA-treated LO2 cells. (**A**) Representative images of intracellular ROS in LO2 cells stained with DCFH-DA. (**B**) Quantification analysis of intracellular ROS. (**C**) Representative images of mitochondria ROS in LO2 cells stained with Mito-SOX. (**D**) Quantification analysis of mitochondrial ROS: cells were treated with 5.5, 11, and 22 μM Empa and 0.5 mM PA for 24 h. (**E**) Representative confocal microscope images of mitochondrial membrane potential by JC-1 staining. (**F**) Quantification analysis of mitochondrial membrane potential by flow cytometry. (**G**) Empa improved intracellular total ATP content impaired by PA (0.5 mM) in LO2 cells. (**H**) Empa improved mitochondrial division induced by PA (0.5 mM). (**I**) MDA content elevated by PA (0.5 mM) was reversed by Empa (5.5, 11 and 22 μM). All data are expressed as mean ± SD (*n* = 3). ** *p* < 0.01 and *** *p* < 0.001 versus Ctrl. # *p* < 0.05, ## *p* < 0.01, and ### *p* < 0.001 versus PA. Ctrl, PA-untreated normal control group; PA, PA-treated control group; PA + Empa 5.5–22, PA-treated and Empa-treated (at final concentrations of 5.5–22 μM) groups.

**Figure 3 antioxidants-11-00799-f003:**
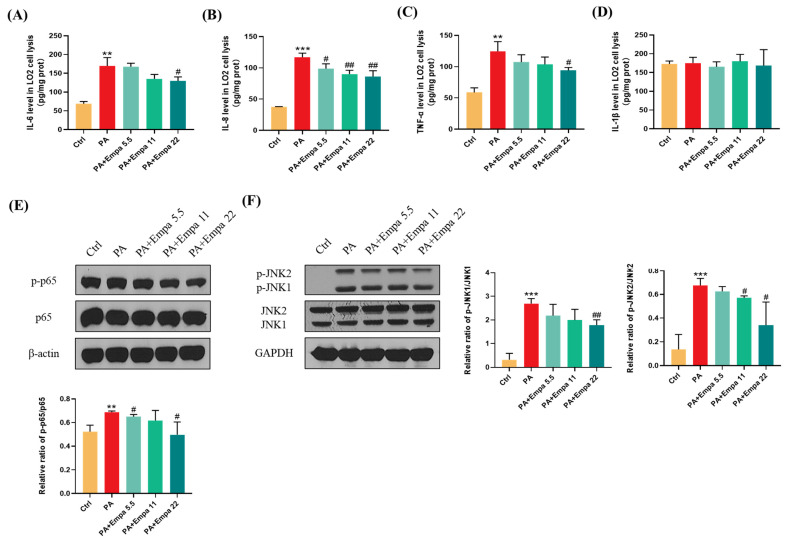
Empa attenuated inflammation level of LO2 cells and inhibited JNK1, JNK2, and NF-κB phosphorylation in a PA-induced lipotoxicity model. (**A**-**D**) Empa decreased intracellular IL-6, IL-8, and TNF-1α protein levels elevated by PA (0.5 mM), but both PA and Empa had no effect on the IL-1β protein level in LO2 cells. (**E**) Empa inhibited *p*-NF-κB level upregulated by PA (0.5 mM) in LO2 cells. (**F**) Empa inhibited *p*-JNK1 and *p*-JNK2 levels elevated by PA (0.5 mM) in LO2 cells. All data are expressed as mean ± SD (*n* = 3). ** *p* < 0.01 and *** *p* < 0.001 versus Ctrl. # *p* < 0.05, and ## *p* < 0.01 versus PA. Ctrl, PA-untreated normal control group; PA, PA-treated control group; PA + Empa 5.5–22, PA-treated and Empa-treated (at final concentrations of 5.5–22 μM) groups.

**Figure 4 antioxidants-11-00799-f004:**
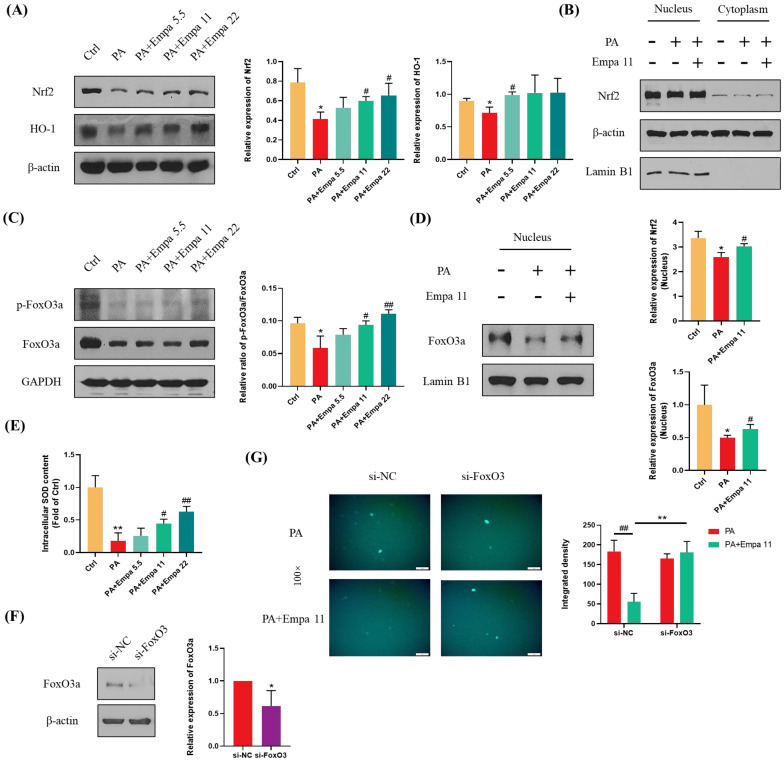
Empa enhanced the cellular antioxidant level of LO2 cells mediated by Nrf2/HO-1 and FoxO3a/SOD pathways. (**A**) Empa increased intracellular Nrf2 and HO-1 protein levels in LO2 cells. (**B**) Empa increased Nrf2 protein level in the nucleus of LO2 cells. (**C**) Empa improved intracellular *p*-FoxO3a protein level impaired by PA (0.5 mM) in LO2 cells. (**D**) Empa promoted FoxO3a expression in the nucleus of LO2 cells. (**E**) Empa improved intracellular total SOD level in LO2 cells. (**F**) FoxO3 knockdown treatment significantly reduced the protein level of FoxO3a in LO2 cells. (**G**) FoxO3 knockdown treatment abolished the improvement of Empa on ROS generation in PA-treated LO2 cells. All data are expressed as mean ± SD (*n* = 3). (**A**–**E**) * *p* < 0.05 and ** *p* < 0.01 versus Ctrl; # *p* < 0.05, and ## *p* < 0.01 versus PA. (**F**–**G**) * *p* < 0.05 and ** *p* < 0.01 versus si-NC; ## *p* < 0.01 versus PA. Ctrl, PA-untreated normal control group; PA, PA-treated control group; PA + Empa 5.5–22, PA-treated and Empa-treated (at final concentrations of 5.5–22 μM) groups.

**Figure 5 antioxidants-11-00799-f005:**
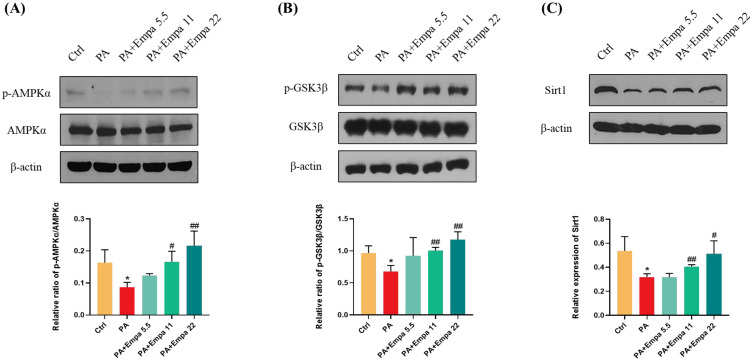
Empa promoted AMPKα and GSK3β phosphorylation and upregulated Sirt1 levels in LO2 cells treated with PA. (**A**) Empa improved intracellular *p*-AMPKα protein level impaired by PA (0.5 mM) in LO2 cells. (**B**) Empa improved intracellular *p*-GSK3β protein level impaired by PA in LO2 cells. (**C**) Empa increased Sirt1 protein level inhibited by PA in LO2 cells. All data are expressed as mean ± SD (*n* = 3). * *p* < 0.05 versus Ctrl. # *p* < 0.05, and ## *p* < 0.01 versus PA. Ctrl, PA-untreated normal control group; PA, PA-treated control group; PA + Empa 5.5–22, PA-treated and Empa-treated (at final concentrations of 5.5–22 μM) groups.

**Figure 6 antioxidants-11-00799-f006:**
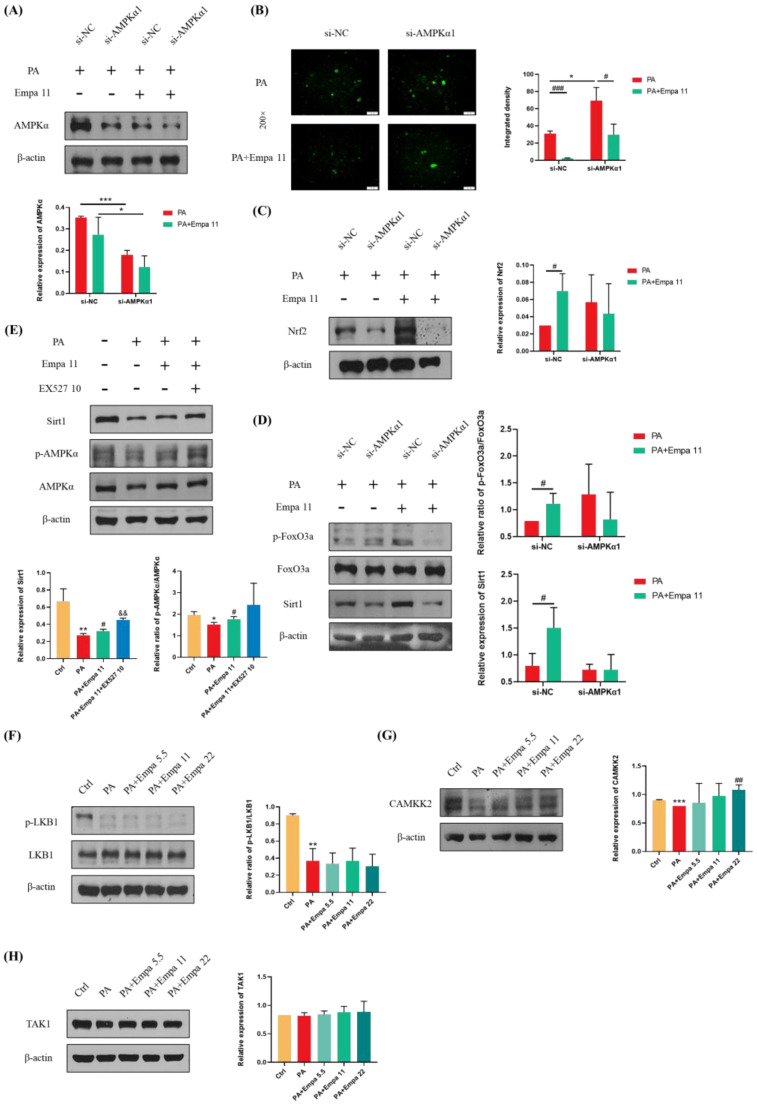
Effects of AMPKα1 on the protective effects of Empa in a PA-induced lipotoxicity in LO2 cell lines and CAMKK2-mediated activation of AMPKα but not phosphorylation of LKB1 or TAK1. (**A**) AMPKα1 knockdown treatment significantly reduced the level of AMPKα in LO2 cells. (**B**) AMPKα1 knockdown treatment enhanced the intracellular ROS generation induced by PA in LO2 cells while inhibiting the ameliorative effect of Empa on the ROS produced by PA. (**C**,**D**) Empa enhanced the expression of Nrf2, Sirt1, and *p*-FoxO3a in PA-treated LO2 cells, but AMPKα1 knockdown treatment partially inhibited this enhancement. (**E**,**F**) Empa did not regulate AMPKα activation through Sirt1 or the phosphorylation of LKB1 in the PA-induced lipotoxicity model. (**G**) CAMKK2 protein level was remarkably upregulated by Empa treatment. (**H**) Neither PA treatment nor PA + Empa treatment had a significant effect on the expression of TAK1. All data are expressed as mean ± SD (*n* = 3). (**A**–**D**) * *p* < 0.05 and *** *p* < 0.001 versus si-NC; # *p* < 0.05 and ### *p* < 0.001 versus PA. (**E**–**H**) * *p* < 0.05, ** *p* < 0.01, and *** *p* < 0.001 versus Ctrl; # *p* < 0.05 and ## *p* < 0.01 versus PA; && *p* < 0.01 versus PA + Empa 11. Ctrl, PA-untreated normal control group; PA, PA-treated control group; PA + Empa 5.5–22, PA-treated and Empa-treated (at final concentrations of 5.5, 11, and 22 μM) groups.

**Figure 7 antioxidants-11-00799-f007:**
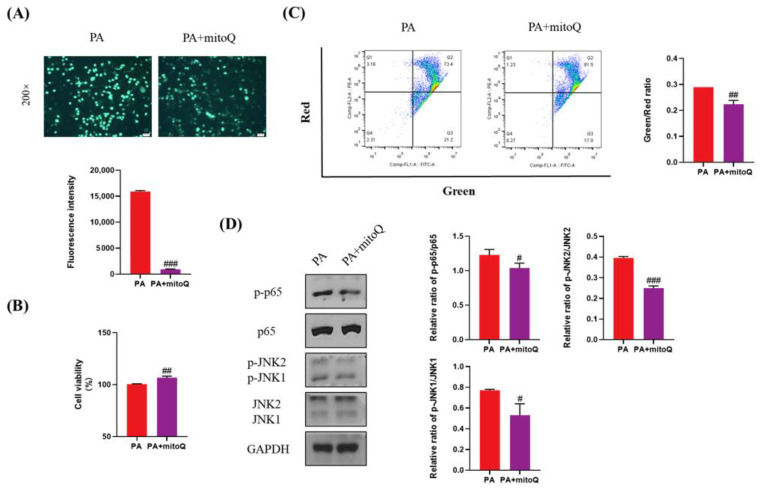
The protective effects of mitoQ on PA-induced lipotoxicity in LO2 cell lines. (**A**) mitoQ (500 nM) treatment strikingly inhibited the ROS generation in PA-treated LO2 cells. (**B**) mitoQ treatment notably improved cell viability in LO2 cells. (**C**) mitoQ treatment ameliorated the loss of mitochondrial membrane potential in LO2 cells. (**D**) mitoQ downregulated the phosphorylation of JNK1, JNK2, and p65 in PA-treated LO2 cells. All data are expressed as mean ± SD. # *p* < 0.05, ## *p* < 0.01, and ### *p* < 0.001 versus PA. PA, PA-treated control group; PA + mitoQ, PA-treated and mitoQ-treated group.

**Figure 8 antioxidants-11-00799-f008:**
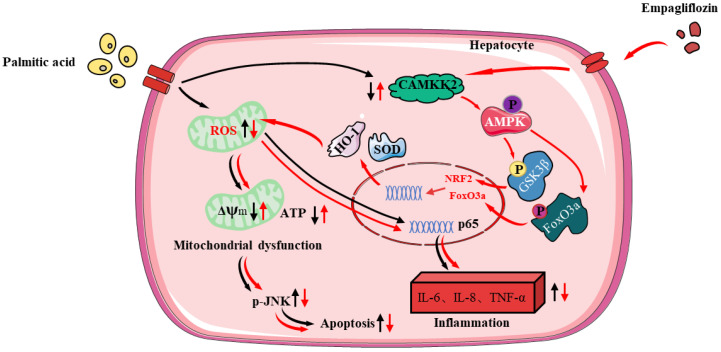
The protective mechanisms of Empa on PA-induced lipotoxicity in LO2 cell lines.

## Data Availability

Data is contained within the article or Appendix A. The data presented in this study are available in [insert article or Appendix A here].

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
