# Peer review of "Empagliflozin-Enhanced Antioxidant Defense Attenuates Lipotoxicity and Protects Hepatocytes by Promoting FoxO3a- and Nrf2-Mediated Nuclear Translocation via the CAMKK2/AMPK Pathway"

_antioxidants, 2022, doi:10.3390/antiox11050799_

Round 1

Reviewer 1 Report

Want et al used an in vitro Palmitic acid-treated LO2 cell line model to minor the nonalcoholic steatohepatitisc and investigated the protective role of Empa in preventing PA-induced cell apoptosis and stress. They found Empa reduced the PA-induced lipocytoxicity, regulated the mitochondrial functions, and restricted the cellular inflammatory response. Finally, they demonstrate the effects of Empa were dependent on AMPK-alpha which potentially acts on FoxO3a to regulate ROS levels. In general, the manuscript is nicely written and experiments are well-designed.

I have a number of concerns for the authors to address, list below:

Mitochondrial electron transport chain is the major source of the cellular ROS. Here lower mitoROS and mitochondrial mass were detected in the Empa group, while the membrane potential was increased. How to explain this discrepancy?

Have the expression of the mitochondrially encoded genes and the mitochondrial DNA copies ever been considered or assayed after Empa treatment?

Figure 2A, 2B and others: the fluorescence intensity was normalized to those of the control, but not to the total cells. Why not depict them as fluorescence+ cell fractions? Does every group have the same number of cells?

Increased FoxO3a phosphorylation was observed in Empa-treated LO2 cells, however, this is insufficient to conclude Empa’s impact on FoxO3a. Given the critical role of FoxO3a in the generation of ROS, A similar knockout experiment using RNAi as did for AMPK-alpha would help to better understand the essential role of FoxO3a in the context of Empa treatment.

AMPK has been reported to increase fatty acid beta-oxidation which may support the TCA cycle and the coupled electron transport chain resulting in mitochondrial activation. However, the authors directed their focus from this aspect to the FoxO3a axis. Has the beta-oxidation pathway ever been evaluated?

A robust experiment would be to perform RNA-sequencing analysis of Empa-treated vs mock-treated LO2 cells in the presence of PA, unbiasedly evaluate the global cellular responses.

Line 62, I believe there is a typo, please correct

Reviewer 2 Report

This is a report investigating the rescue of PA lipotoxicity by Empa. Although the data are abundant (an extensive study of the possibly involved pathways) and appear to be able to reach a conclusion, I have the following concern:

Major: 

This study suffered from a fundamental design flaw, which is the lack of study of Empa alone on various parameters. If Empa alone could decrease or increase some of those parameters, then it is very difficult to interpret the PA+Empa groups, ie., the "modulation" may not be a consequence of the effects of Empa on PA.

In particular, one can see that the rescue effect of Empa on cell viability is very slim. How do the authors negate the possibility that Empa itself caused a slight enhancement on cell proliferation?

Minor:

1. I would usually prefer to see drug conc. in uM instead of ug/ml.

2. The # signs to indicate stat significance are always put on the control groups. I believe it is better to put it in the PA group, and then another sign to indicate any significant difference between PA+Empa and PA alone.

Round 2

Reviewer 1 Report

Line 349: FoxO3a was involved

Line 352: Further, FoxO3 knockdown abolished

Reviewer 2 Report

The authors only carried out limited additional expt as control, as shown in supplement. The authors need to check the effects of Empa itself (choose the highest conc.) on most of the parameters. These control results must be well presented.

The lack of time to do additional expt is not an excuse, and I presume the Editor should be willing to provide more time for revision. Scientific rigor must be ensured to qualify publication.

In addition, in the entire MS including the figures, the unit should be uM instead ug/ml, which has to be corrected.
